# SELF-IMPROVING MEDICAL VISUAL QUESTION ANSWERING THROUGH REASONING TRAJECTORY CLUSTERING

## ABSTRACT

While large language models have shown promise in medical applications, their performance in medical visual question answering (VQA) remains limited by insufficient vision-language reasoning capabilities. We address this challenge through two complementary approaches. First, we generate high-quality reasoning annotations for existing medical VQA datasets using COMCTS algorithm. Second, we introduce a self-improvement framework that bootstraps model performance by learning from its own outputs, guided by a small set of high-quality reasoning samples. To optimize this self-improvement process, we propose a novel filtering mechanism based on reasoning trajectory K-medoids clustering, which employs Dynamic Time Warping (DTW) distances to select the most effective generated reasoning paths. Our comprehensive approach demonstrates significant improvements in medical VQA tasks. We release both the COMCTS-generated reasoning datasets and our code to support future research. Our code is available at https://anonymous.4open.science/r/SelfImproving-MedicalVQA-5507

## 1 INTRODUCTION

Reasoning in large language models has been extensively studied in recent years. Researchers have proposed models with strong reasoning capabilities, which are particularly important for tasks such as mathematical problem solving and medical image interpretation (Ahn et al., 2024; Shao et al., 2024; Chen et al., 2024a; Imani et al., 2023)

Recent work has focused on developing vision-language models for medical visual question answering and report generation (Wu et al.; Lin et al., 2023; Tu et al., 2024; Zhang et al., 2023a;b). However, progress in this area is often limited by the scarcity of comprehensive medical datasets. As a result, many studies rely on either simple question-answer pairs (e.g., 'Which organ is presented in this image?') or noisy image-caption pairs extracted from publications or the web.

Despite the advances in large language models, reasoning in medical visual question answering remains an underexplored area (Liévin et al., 2024; Kwon et al., 2024). One contributing factor is the lack of vision-language datasets that include detailed rationales. Most existing datasets used for medical visual question answering provide only direct answers, with a significant portion limited to yes/no responses (Lau et al., 2018; He et al., 2020; Ben Abacha et al., 2019; Liu et al., 2021; Hu et al., 2024; Zhang et al., 2023b; Lozano et al., 2024)

To address these challenges, this paper makes the following contributions:

- We generate rationales for the VQA-RAD, SLAKE-VQA, PathVQA, PMC-VQA, and OmniMed-VQA datasets using the COMCTS algorithm (Yao et al., 2024). This generation process involves

Qwen2-VL-7B and Gemma-3-27B models, with the DeepSeek-R1 model used for rationale evaluation. Due to the high computational cost, we limit the number of samples in this step.

- We introduce a self-improvement framework for medical visual question answering. After training our models on rationale-augmented data, we run inference on samples that contain only ground-truth answers (i.e., without rationales). The generated answers are filtered using an LLM API to retain only those that match the ground-truth responses.

- We further enhance the self-improvement process by incorporating reasoning-based filtering. Specifically, we compare the model-generated rationales to the original ones using K-medoids clustering with DTW distance. If a generated rationale is sufficiently close to one of the center of the clusters, it is retained; otherwise, it is discarded. We observe that this clustering-based filtering improves performance more effectively than filtering based solely on answer correctness.

## 2 RELATED WORK

Medical vision-and-language understanding has received increasing attention in recent years. (Wu et al., 2024) proposed a LLaMA-based model specifically designed for multiple-choice medical question answering. LLaVA-Med, introduced by (Li et al., 2023), is a conversation-based medical AI system. PMC-CLIP, trained on PMC data, is presented in (Lin et al., 2023). R-LLaVA (Chen et al., 2024b) enhances medical visual question answering by incorporating visual regions of interest.

(Zelikman et al., 2024) proposed Self-Taught Reasoner (STaR), which generates rationale–answer pairs and incorporates these pairs—along with the original question—into the fine-tuning process, provided the generated answer is correct. (Hosseini et al., 2024) introduced V-STaR, which includes a generator and a verifier model. The generator is trained in a supervised manner on its own correctly generated answers, while the verifier is updated using DPO on both correct and incorrect outputs from the generator.

Similarly, (Pang et al., 2024) presented a framework in which a language model produces chains of thought (CoTs) and answers from prompts. These CoT–answer pairs are scored by a reward model, and the best-performing pairs are used to update the language model using a combination of DPO and negative log-likelihood (NLL) losses.

ReST, introduced by (Gulcehre et al., 2023), follows a two-phase process: a "Grow" step in which the policy generates data, and an "Improve" step in which a filtered subset of that data is used to update the policy. Our work closely aligns with ReST, but introduces a specific filtering mechanism that leverages the structure of reasoning trajectories.

Several techniques have been proposed for generating datasets and chains of thought (CoTs). The Chain of Thought (CoT) approach is among the most widely used methods for improving reasoning in language models (Wei et al., 2022). (Wang et al., 2022) introduced the concept of self-consistency in CoT generation, which enhances reasoning performance by selecting the most consistent reasoning paths through majority voting on final answers.

Tree of Thoughts (ToT), proposed by (Yao et al., 2023), extends this idea by generating reasoning paths in a structured tree format—using breadth-first or depth-first search—rather than producing independent chains. (Lample et al., 2022) proposed a tree-based approach for automated theorem proving.

More recently, (Yao et al., 2024) introduced a collective Monte Carlo Tree Search (COMCTS) method for generating reasoning paths. We adopt this approach in our dataset generation phase due to its effectiveness in exploring diverse and high-quality reasoning trajectories.

# 3 METHOD

Our method consists of two main phases: (1) constructing the reasoning and self-improvement datasets, and (2) pretraining models on these datasets followed by supervised fine-tuning on the original data. The algorithms corresponding to these phases are detailed in Algorithm 1, Algorithm 2, and illustrated in Figure 1.

We begin our method with an original visual question answering dataset, such as Slake-VQA, denoted by $\mathcal{D}_{VQA}$. It is common for VQA datasets—particularly in the medical domain—to contain short, direct answers without accompanying chains of thought or detailed reasoning in their ground truth annotations. To address this, we employ the COMCTS algorithm to generate a reasoning dataset that includes both chains of thought and final answers.

---

**Algorithm 1:** Dataset Construction Pipeline: COMCTS + Self-Improvement

**Input:** Original VQA dataset $\mathcal{D}_{VQA}$
**Output:** Reasoning dataset $\mathcal{D}_{reas}$; Self-improvement datasets $\mathcal{F}_{ca}(\mathcal{D}_{SI})$, $\mathcal{F}_{ca+cl}(\mathcal{D}_{SI})$

1. Apply the COMCTS algorithm to a small subset of $\mathcal{D}_{VQA}$ to generate a reasoning-augmented dataset $\mathcal{D}_{reas}$.

2. Train a vision-language model $\mathcal{M}_{reas}(\cdot)$ on $\mathcal{D}_{reas}$.

3. Use $\mathcal{M}_{reas}(\cdot)$ to perform inference on the remaining samples in $\mathcal{D}_{VQA}$ to obtain a self-improvement dataset $\mathcal{D}_{SI}$.

4. Filter $\mathcal{D}_{SI}$ using two strategies:

   - **Correct Answer (CA) Filtering**: Retain samples with correct answers $\rightarrow \mathcal{F}_{ca}(\mathcal{D}_{SI})$
   - **Correct Answer + Clustering (CA+CL) Filtering**: Further filter samples by comparing generated reasoning trajectories to originals using K-medoids clustering $\rightarrow \mathcal{F}_{ca+cl}(\mathcal{D}_{SI})$

---

The details of the Algorithm 1 are provided in Section 3.1 and 3.2. In the Algorithm 1, we not only utilize COMCTS for reasoning dataset creation $\mathcal{D}_{reas}$ but also run self-improvement part to generate self-improvement dataset $\mathcal{D}_{SI}$ and its filtered versions $\mathcal{F}_{ca}(\mathcal{D}_{SI})$, $\mathcal{F}_{ca+cl}(\mathcal{D}_{SI})$

## 3.1 REASONING-DATASET GENERATION WITH COMCTS

COMCTS (Yao et al., 2024) is a method for generating reasoning paths based on questions using multiple large language models. These models can iteratively build upon each other's thoughts, generating successive reasoning steps. At each step, the generated reasoning paths are evaluated by an additional language model via an API provided by the original authors, allowing the elimination of problematic or irrelevant thoughts within the reasoning tree. This process continues until a final correct answer is obtained or a maximum iteration limit is reached. For further technical details, we refer readers to the original paper.

In our experiments, we utilize the Gemma-3-27B and Qwen2-VL-7B models to generate reasoning paths, and the DeepSeek model to verify the correctness of the thoughts (Team et al., 2025; Wang et al., 2024; Guo et al., 2025). We denote the resulting dataset, which includes both reasoning chains and detailed answers, as $\mathcal{D}_{reas}$

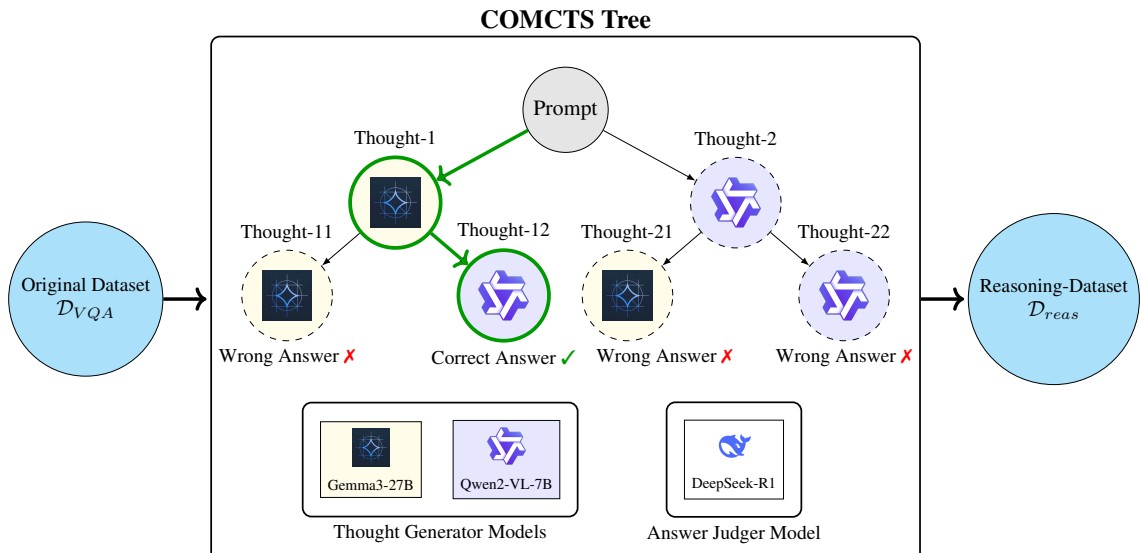

We treat as $\mathcal{D}_{reas}$ a ground truth reasoning dataset in the self-improvement phase. This is justified by the fact that COMCTS leverages larger and more powerful vision-language models than those we train during self-improvement, combined with iterative correctness checks. Therefore, the generated reasoning paths are highly likely to be meaningful and reliable.

## 3.2 SELF-IMPROVEMENT

Our goal is to leverage the reasoning dataset to generate an improved reasoning dataset through a self-improvement or offline reinforcement learning approach. Specifically, we first train our vision-language model $\mathcal{M}(\cdot)$ on the reasoning dataset $\mathcal{D}_{reas}$, resulting in a trained model $\mathcal{M}_{reas}(\cdot)$. We then perform inference with $\mathcal{M}_{reas}(\cdot)$ on samples from the original dataset $\mathcal{D}_{VQA}$ that were not included in $\mathcal{D}_{reas}$, with the expectation that the model will produce meaningful rationales along with correct answers for the given image-question pairs. We refer to this newly generated dataset as $\mathcal{D}_{SI}$, where $SI$ stands for self-improvement.

However, some of the generated rationales may contain incorrect answers or exhibit meaningless or contradictory chains of thought. To address this, we apply a filtering process. The first filter discards samples with incorrect final answers, retaining only those with correct answers; we denote this filtered set as $\mathcal{F}_{ca}(\mathcal{D}_{SI})$ where $ca$ stands for correct-answer. The correctness of answers is verified using a separate large language model API, which compares the generated final answers with the ground truth from the original dataset $\mathcal{D}_{VQA}$.

Additionally, we apply a more detailed filtering by examining the reasoning chains themselves, resulting in the filtered set $\mathcal{F}_{ca+cl}(\mathcal{D}_{SI})$ where $ca + cl$ stands for correct-answer + clustering . We now proceed to describe these filtering approaches in detail.

## 3.3 FILTERING

The first filtering step checks whether the generated final answer is correct. Specifically, we extract the final answer from the rationales produced by our model in $\mathcal{D}_{SI}$ and compare it with the ground truth answer in $\mathcal{D}_{VQA}$. This comparison is handled by an LLM API. If the LLM determines that the final answer is incorrect,

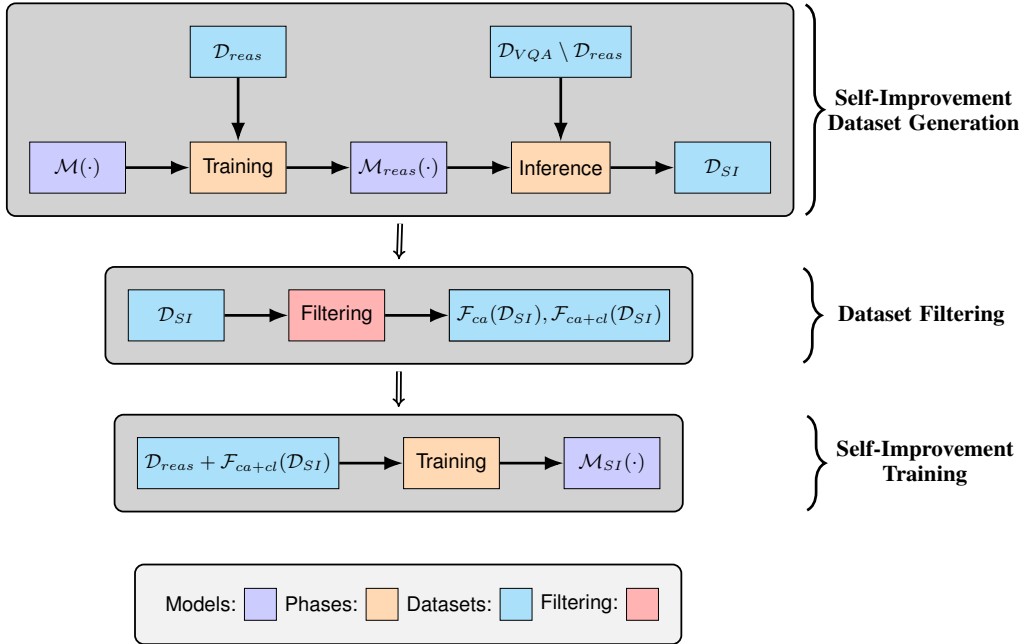

Figure 1: Overview of the self-improvement process.

we discard the corresponding sample. We denote the resulting filtered dataset, which only includes samples with correct answers, as $\mathcal{F}_{ca}(\mathcal{D}_{SI})$. In the next section, we describe our clustering-based filtering approach, which results in the dataset $\mathcal{F}_{ca+cl}(\mathcal{D}_{SI})$.

Now, consider a sample from the reasoning dataset (image, question, answer) $\in \mathcal{D}_{reas}$. We break down the *answer* into a sequence of sentences, denoted as (thought-1, thought-2, ..., thought-m), where thought-j is the $j^{\text{th}}$ sentence in the rationale.

We obtain the representations of both the input image and the individual thoughts using pretrained models. For image features, we use the facebook/dino-vits16 model (Caron et al., 2021), and for sentence representations, we use the all-MiniLM-L6-v2 model from Sentence Transformers (Reimers & Gurevych, 2019). As a result, we construct the following sequence of vectors: $\mathcal{S} = (\mathbf{r}_0, \mathbf{r}_1, \mathbf{r}_2, \ldots, \mathbf{r}_m)$, where $\mathbf{r}_0 \in \mathbb{R}^d$ corresponds to the image features and $\mathbf{r}_j \in \mathbb{R}^d$ represents the features of thought-$j$ for $j = 1, 2, \ldots, m$. Given our choice of models, the feature dimension is $d = 384$.

### 3.3.1 DTW DISTANCE BETWEEN TRAJECTORIES

Next, we analyze the similarity between the reasoning trajectories in our generated dataset $\mathcal{D}_{SI}$ and those in the original reasoning dataset $\mathcal{D}_{reas}$. To do this, we define a metric that captures the difference between two sequences of vector representations. Specifically, we use Dynamic Time Warping (DTW) as our similarity metric (Müller, 2007).

Assume we are given two reasoning trajectories:

$$\mathcal{S}_i = (\mathbf{r}_{i0}, \mathbf{r}_{i1}, \mathbf{r}_{i2}, ..., \mathbf{r}_{im}) \text{ and } \mathcal{S}_j = (\mathbf{r}_{j0}, \mathbf{r}_{j1}, \mathbf{r}_{j2}, ..., \mathbf{r}_{jk})$$

**Question: Does the heart appear normal size?     Ground Truth Answer: No**

$\mathbf{r}_1$ : The image is a grayscale chest X-ray.

$\mathbf{r}_2$ : It shows the patient's lungs, heart, and ribs.,

$\mathbf{r}_3$ : The lungs appear clear in the upper regions, but there are some areas of increased density in the lower lung fields bilaterally.

$\mathbf{r}_4$ : The heart is visible in the center of the image, and appears somewhat enlarged.

$\mathbf{r}_5$ : The bony structures of the thorax are also visible.

$\mathbf{r}_6$ : The question asks about the size of the heart as visualized in the X-ray.

$\mathbf{r}_7$ : We need to compare the heart size to the overall width of the thorax.

$\mathbf{r}_8$ : A normal heart size should occupy less than half the width of the chest at the widest point.

$\mathbf{r}_9$ : The appearance of an enlarged heart (cardiomegaly) can indicate various cardiac conditions.

$\mathbf{r}_{10}$ : The final answer is: No.

Figure 2: The sentence features are extracted by a Sentence Transformer (all-MiniLM-L6-v2) and represented by $\mathbf{r}_j \in \mathbb{R}^d$ for $j = 1, \ldots, 10$. The image features are extracted by a Vision Transformer (facebook/dino-vits16) as $\mathbf{r}_0 \in \mathbb{R}^d$. The final trajectory of this sample is $\mathcal{S} = (\mathbf{r}_0, \mathbf{r}_1, \ldots, \mathbf{r}_{10})$.

where all vectors $\mathbf{r}_{i0}, \mathbf{r}_{i1}, \ldots, \mathbf{r}_{im}, \mathbf{r}_{j0}, \mathbf{r}_{j1}, \ldots, \mathbf{r}_{jk} \in \mathbb{R}^d$. To properly define the DTW metric, we first introduce the concept of a warping path and its associated total cost. We follow the notation presented in (Müller, 2007).

**Definition 1.** *A (m,k)-warping path is a sequence $p = (p_0, ..., p_L)$ with $p_l = (m_l, k_l) \in [0 : m] \times [0 : k]$ for $l \in [1 : L]$ satisfying the following three conditions.*
*(i) Boundary condition: $p_0 = (0, 0)$ and $p_L = (m, k)$.*
*(ii) Monotonicity condition: $m_1 \le m_2 \le ... \le m_L$ and $k_1 \le k_2 \le ... \le k_L$.*
*(iii) Step size condition: $p_{l+1} - p_l \in \{(1, 0), (0, 1), (1, 1)\}$ for $l \in [0 : L - 1]$.*

**Definition 2.** *The total cost of a warping-path $p = (p_0, ..., p_L)$ is defined as*

$$c_p(\mathcal{S}_i, \mathcal{S}_j) := \sum_{i=0}^{L} c(\mathbf{r}_{im_l}, \mathbf{r}_{jm_l}).$$

*where the choice of cost function $c$ is user-defined.*

**Definition 3.** *The DTW distance between two trajectories is given as*

$$DTW(\mathcal{S}_i, \mathcal{S}_j) := c_p^*(\mathcal{S}_i, \mathcal{S}_j) = min\{c_p(\mathcal{S}_i, \mathcal{S}_j) | p \text{ is an } (m, k) - warping - path\}$$

We define the cost function $c$ as the Euclidean distance. Henceforth, we use the DTW distance to compute the similarity between two rationale trajectories.

### 3.3.2 CLUSTERING ON THE TRAJECTORIES OF THE REASONING DATASET $\mathcal{D}_{reas}$

We apply a clustering method to all reasoning trajectories in $\mathcal{D}_{reas}$. Then, for each trajectory in the candidate reasoning dataset $\mathcal{F}_{ca}(\mathcal{D}_{SI})$, we compute its distance to the closest cluster identified in $\mathcal{D}_{reas}$.

Since we can compute pairwise distances between trajectories—even though they may have varying lengths—we use the K-Medoids clustering algorithm (Kaufman & Rousseeuw, 2009), which is well-suited for this setting.

$$\mathcal{D}_{SI} \xrightarrow{\textit{Correct Answers Only}} \mathcal{F}_{ca}(\mathcal{D}_{SI}) \xrightarrow{\textit{Clustering Based Filtering}} \mathcal{F}_{ca+cl}(\mathcal{D}_{SI})$$

We first apply K-Medoids clustering to the reasoning dataset $\mathcal{D}_{reas}$ with $K = 10$. Let the resulting cluster centers be denoted as $C_1, C_2, \ldots, C10$. For each sample in $\mathcal{F}_{ca}(\mathcal{D}_{SI})$, we compute its distance to all cluster centers and record the minimum. We then sort the candidate reasoning paths based on their distance to the nearest cluster and discard the top $p$ fraction with the highest distances ($p = 0.2$ in our experiments). This gives us the final filtered dataset, denoted as $\mathcal{F}_{ca+cl}(\mathcal{D}_{SI})$.

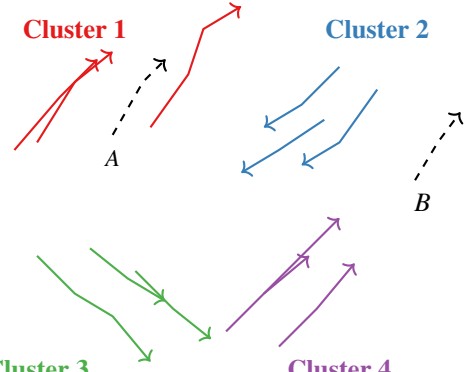

Figure 3: We apply K-medoids clustering to the trajectories of the reasining dataset $\mathcal{D}_{reas}$ which is plotted in red, blue, green, purple. Also there are two new trajectories A and B with dashed lines which represent candidate trajectories from the $\mathcal{F}_{ca}(\mathcal{D}_{SI})$ dataset. As it can be deduced from the figure, it can be expected that A belongs to the Cluster 1, while B does not seem to belong any of the clusters above, which can be considered as an outlier and discarded. In our experiments we discard 20% of the all candidate trajectories with highest distances to the centers of the clusters.

## 3.4 SFT ON $\mathcal{D}_{VQA}$ SELF-IMPROVED MODEL TESTING

---
**Algorithm 2:** Evaluation of Training Strategies on the Original VQA Dataset

---
**Input:** Datasets: Original VQA dataset $\mathcal{D}_{VQA}$; Self-improvement datasets $\mathcal{F}_{ca}(\mathcal{D}_{SI})$, $\mathcal{F}_{ca+cl}(\mathcal{D}_{SI})$
**Output:** Trained models: $\mathcal{M}_{base}$, $\mathcal{M}_{reas}$, $\mathcal{M}_{SI-ca}$, $\mathcal{M}_{SI-ca+cl}$

1. **Baseline**: Train and evaluate a model solely on $\mathcal{D}_{VQA}$ to obtain the baseline model $\mathcal{M}_{base}$.

2. **Reas**: Train on the reasoning-augmented dataset $\mathcal{D}_{reas}$ and perform supervised fine-tuning (SFT) on $\mathcal{D}_{VQA} \rightarrow \mathcal{M}_{reas}$.

3. **SI-ca**: Train on $\mathcal{D}_{reas} \cup \mathcal{F}_{ca}(\mathcal{D}_{SI})$ and perform SFT on $\mathcal{D}_{VQA} \rightarrow \mathcal{M}_{SI-ca}$.

4. **SI-ca+cl**: Train on $\mathcal{D}_{reas} \cup \mathcal{F}_{ca+cl}(\mathcal{D}_{SI})$ and perform SFT on $\mathcal{D}_{VQA} \rightarrow \mathcal{M}_{SI-ca+cl}$.

---

Finally, we curated our self-generated dataset and are ready to train a model from scratch using $\mathcal{D}_{reas}$ along with the filtered versions of $\mathcal{D}_{SI}$. To evaluate the impact of our self-improvement process, we train models on two combinations: $\mathcal{D}_{reas} + \mathcal{F}_{ca}(\mathcal{D}_{SI})$ and $\mathcal{D}_{reas} + \mathcal{F}_{ca+cl}(\mathcal{D}_{SI})$. Afterwards, these models are fine-tuned on $\mathcal{D}_{VQA}$ to verify whether the self-improvement step actually enhances performance on the original task. The details of the methods used in the experiments are provided in Section 4.2.

## 4 EXPERIMENTS

### 4.1 DATASETS

In our experiments, we use the VQA-RAD, Slake-VQA, Path-VQA, PMC-VQA, and Omnimed-VQA datasets. As shown in Table A.1, we do not use the full set of samples from these datasets. Instead, we randomly select a subset from each to construct the reasoning dataset $\mathcal{D}_{reas}$ using COMCTS. Another random subset is used in the self-improvement stage, forming $\mathcal{D}_{SI}$. We also restrict $\mathcal{D}_{VQA}$ to include only the samples that appear in either $\mathcal{D}_{reas}$ or $\mathcal{D}_{SI}$. Since the total number of samples in these datasets is large and both COMCTS and the self-improvement process are computationally expensive, we limit the number of samples selected from each dataset which is provided in section A.1.

### 4.2 METHODS

$\mathcal{M}_{base}$: We train and evaluate our models solely on the original dataset $\mathcal{D}_{VQA}$, without any form of pretraining.

$\mathcal{M}_{reas}$: We first train our models on the reasoning dataset $\mathcal{D}_{reas}$, followed by supervised fine-tuning on $\mathcal{D}_{VQA}$. We then evaluate the models on $\mathcal{D}_{VQA}$.

$\mathcal{M}_{SI-ca}$: We first train our models on the reasoning dataset $\mathcal{D}_{reas}$. Next, we perform inference on the remaining samples in $\mathcal{D}_{VQA} \setminus \mathcal{D}_{reas}$, which we denote as $\mathcal{D}_{SI}$. In this step, we generate reasoning trajectories for the samples that lack explanatory rationales. We then continue training the model on the combined dataset $\mathcal{D}_{VQA} + \mathcal{F}_{ca}(\mathcal{D}_{SI})$, followed by supervised fine-tuning on $\mathcal{D}_{VQA}$.

$\mathcal{M}_{SI-ca+cl}$: We start by training our models on the reasoning dataset $\mathcal{D}_{reas}$. Then, we run inference on the remaining samples in $\mathcal{D}_{VQA} \setminus \mathcal{D}_{reas}$, denoted as $\mathcal{D}_{SI}$, where we generate reasoning trajectories for samples that do not include explanations. We further filter $\mathcal{D}_{SI}$ using K-Medoids clustering based on DTW distances between reasoning trajectories. The model is then trained on the combined dataset $\mathcal{D}_{VQA} + \mathcal{F}_{ca+cl}(\mathcal{D}_{SI})$, followed by supervised fine-tuning on $\mathcal{D}_{VQA}$.

### 4.3 RESULTS

| Model | Dataset | $\mathcal{M}_{base}$ | $\mathcal{M}_{reas}$ | $\mathcal{M}_{SI-ca}$ | $\mathcal{M}_{SI-ca+cl}$ |
|---|---|---|---|---|---|
| ViT-B/16 + DS-R1-Llama-8B | VQA-RAD | 37.82 | 41.76 | 41.53 | **42.46** |
| ViT-B/16 + DS-R1-Qwen-1.5B | VQA-RAD | 35.27 | **35.5** | **35.5** | 33.87 |
| ViT-B/16 + DS-R1-Llama-8B | Slake-VQA | 68.0 | 67.2 | 65.2 | **70.8** |
| ViT-B/16 + DS-R1-Qwen-1.5B | Slake-VQA | 65.0 | 65.6 | 67.4 | **67.6** |
| ViT-B/16 + DS-R1-Llama-8B | Path-VQA | **50.6** | 48.2 | 49.4 | 47.2 |
| ViT-B/16 + DS-R1-Qwen-1.5B | Path-VQA | 46.8 | 47.0 | **47.4** | 46.8 |
| ViT-B/16 + DS-R1-Llama-8B | PMC-VQA | **5.6** | 4.4 | 3.4 | 3.6 |
| ViT-B/16 + DS-R1-Qwen-1.5B | PMC-VQA | 2.4 | 3.8 | 4.0 | **5.0** |
| ViT-B/16 + DS-R1-Llama-8B | Omnimed-VQA | 46.2 | 47.2 | 45.8 | **47.4** |
| ViT-B/16 + DS-R1-Qwen-1.5B | Omnimed-VQA | 38.0 | 35.4 | 38.6 | **41.2** |
| Mean | | 39.57 | 39.61 | 39.82 | **40.59 (Ours)** |
| Best (Count) | | 2/10 | 1/10 | 2/10 | **6/10 (Ours)** |

Table 1: Accuracy values for each original dataset $\mathcal{D}_{VQA}$ are provided. As we can see from the mean accuracies, the performance improves if we add the reasoning dataset in the pretraining dataset, namely $\mathcal{M}_{base} < \mathcal{M}_{reas}$ in terms of the performance. We also observe that including self-improvement with correct answers can further improve the performance so we get $\mathcal{M}_{reas} < \mathcal{M}_{SI-ca}$. Finally, adding our clustering method improves the performance further and we get $\mathcal{M}_{SI-ca} < \mathcal{M}_{SI-ca+cl}$ in terms of accuracy scores.

As shown in Table 4.3 and A.3, our approach outperforms the baseline model $\mathcal{M}_{base}$, which is trained only on the original dataset $\mathcal{D}_{VQA}$ without any pretraining. Moreover, our method achieves higher accuracy than the model trained solely on the reasoning dataset $\mathcal{D}_{reas}$, without any self-improvement steps (denoted as $\mathcal{M}_{reas}$). During the self-improvement phase, we also observe that filtering out incorrect answers using an LLM API (denoted $\mathcal{M}_{SI-ca}$) provides noticeable gains. However, additional improvement is obtained when K-Medoids clustering is applied to the generated reasoning trajectories. This clustering step further reduces noisy or low-quality rationales, resulting in stronger performance on the original task, denoted as $\mathcal{M}_{SI-ca+cl}$. All reported accuracy scores are computed using exact string match between the predicted and ground-truth answers.

We also report BLEU-1 scores for our methods. The motivation for including BLEU-1 is that models may generate answers that are semantically close to the ground truth but not exactly identical, which can lead to underestimating performance when using strict accuracy. BLEU-1 is better suited to capture such near matches. For instance, as shown in the results, the naive accuracy on the PMC-VQA dataset appears quite low, but BLEU-1 provides a more nuanced evaluation of model performance. We report both sentence-level and corpus-level BLEU-1 scores. Consistent with the accuracy metrics, our method also outperforms the baselines in both sentence-level and corpus-level BLEU-1 scores.

| Model | Dataset | $\mathcal{M}_{base}$ | $\mathcal{M}_{reas}$ | $\mathcal{M}_{SI-ca}$ | $\mathcal{M}_{SI-ca+cl}$ |
|---|---|---|---|---|---|
| ViT-B/16 + DS-R1-Llama-8B | VQA-RAD | 56.55 | **60.65** | 59.03 | 59.0 |
| ViT-B/16 + DS-R1-Qwen-1.5B | VQA-RAD | 52.93 | 52.35 | 53.34 | **53.39** |
| ViT-B/16 + DS-R1-Llama-8B | Slake-VQA | 73.07 | 71.94 | 71.32 | **74.86** |
| ViT-B/16 + DS-R1-Qwen-1.5B | Slake-VQA | 70.48 | 70.82 | 71.59 | **72.11** |
| ViT-B/16 + DS-R1-Llama-8B | Path-VQA | **68.77** | 66.13 | 66.41 | 65.04 |
| ViT-B/16 + DS-R1-Qwen-1.5B | Path-VQA | 65.93 | 66.69 | 67.0 | **67.99** |
| ViT-B/16 + DS-R1-Llama-8B | PMC-VQA | **47.0** | 45.3 | 44.89 | 46.47 |
| ViT-B/16 + DS-R1-Qwen-1.5B | PMC-VQA | 42.76 | 46.25 | 44.79 | **46.49** |
| ViT-B/16 + DS-R1-Llama-8B | Omnimed-VQA | 59.77 | 61.3 | 59.67 | **61.85** |
| ViT-B/16 + DS-R1-Qwen-1.5B | Omnimed-VQA | 55.78 | 55.6 | 55.85 | **59.43** |
| Mean | | 59.3 | 59.7 | 59.39 | **60.66 (Ours)** |
| Best (Count) | | 2/10 | 1/10 | 0/10 | **7/10 (Ours)** |

Table 2: We report sentence-level BLEU-1 scores. As shown by the average scores, applying self-improvement based only on final answer correctness does not lead to better performance—i.e., $\mathcal{M}_{reas} > \mathcal{M}_{SI-ca}$. However, when we add clustering on top of the correct-answer filtering, we observe the best performance

## 5 CONCLUSION

In this work, we present a method for generating medical reasoning datasets using COMCTS (Yao et al., 2024), and we release our generated dataset to support future research. In addition, we introduce a self-improvement framework that enhances dataset quality by filtering reasoning paths based on both answer correctness and clustering with K-Medoids, using Dynamic Time Warping (DTW) distances between reasoning trajectories. Our experiments show that this combination significantly improves model performance on downstream tasks.

We leave a deeper investigation of reasoning trajectories—beyond their use in self-improvement—as future work. Analyzing reasoning sequences through DTW-based similarity can be useful not only for building higher-quality datasets and removing noisy examples (as we demonstrated), but also for identifying creative or novel reasoning paths. Such trajectories may appear as outliers but still offer valid, insightful perspectives that are not commonly found in the dataset.

