# OpenReview forum: "Self-Improving Medical Visual Question Answering through Reasoning Trajectory Clustering"
_ICLR.cc/2026/Conference — Submitted to ICLR 2026_

### Official Review · Reviewer_Cb2n · 2025-10-23

**Soundness:** 2
**Presentation:** 2
**Contribution:** 2
**Rating:** 4
**Confidence:** 4

**Summary:**

The paper targets vision–language reasoning in medical VQA. It first augments existing medical VQA datasets with chain-of-thought rationales generated via COMCTS using Qwen2-VL-7B and Gemma-3-27B, with DeepSeek-R1 for rationale scoring. It then proposes a self-improvement framework: train on rationale-augmented data, infer on answer-only samples, and keep generations that match ground truth; further, it filters by comparing generated reasoning traces to original ones using K-medoids with DTW distance. The authors report gains and release code and rationale data.

**Strengths:**

- The code is open-sourced.
- The reasoning data over medical datasets might be valuable to the community.

**Weaknesses:**

From my perspective, the current manuscript is more like an engineering report that composes existing methods: COMCTS + open-source LLMs (Qwen2-VL/Gemma-3/Deepseek-R1) + DTW-based K-medoids. The novelty might be limited and largely in the specific domain rather than a new learning principle. Additionally, it is unclear the improvement comes from more reasoning data or just more data.

**Questions:**

## Baselines and more ablation studies
1. There are several methods for bootstrapping reasoning in general LLMs: STaR, V-STaR. The authors should consider to compare with previous approaches. Additionally, I recommend the authors to clearly articulate the conceptual advance beyong combining COMCTS + DTW-based K-medoids.

> L219-221: For image features, we use the facebook/dino-vits16 model (Caron et al., 2021), and for sentence representations, we use the all-MiniLM-L6-v2 model from Sentence Transformers

2. Why DTW + K-medoids? Why dino-vits and MiniLM are chosen? Is there any ablation study? It might be better if the authors could justify these choices with sensitivity analyses (different hyper-parameters, alternative distances, alternative clustering).

2.1 Does self-consistency majority vote work here?

3. Do DTW distances actually correlate with human-rated rationale quality in medical research?

## Reliance on LLM
4. It seems that the construction of reasoning datasets relys on LLMs generally. Is there any hallucination? Are the reasoning traces verified by medical experts?

5. The authors should report prompts, calibration, and exact rules; quantify false accepts/rejects. Additionally, is there any leakage risk if the judge saw the same datasets during pretraining?

## Experiments

6. Mean accuracy moves from 39.57 -> 40.59 .

    6.1 Is this (less than 3% improvement) signifcant?

    6.2 How to distinguish the performance improvement from more data / more reasoning traces? The authors should provide more explanations. Additionally, the authors should provide more details about confidence intervals, multiple seeds, and standardized compute.

> Table 2: We report sentence-level BLEU-1 scores

7. Is BLEU-1 suitable for short or medical VQA answers? The authors should include per-type breakdowns (yes/no vs open-ended), and clinical-error criticality analysis.

8. Report training/inference budgets for COMCTS, judge calls, and clustering; provide full prompts and code to reproduce DTW pipelines and clustering seeds; quantify cost vs. accuracy trade-offs. It might be better if the authors could model how performance scales with the amount of COMCTS data.

## Typos

Typos: Figure 3, "the reasining dataset" -> Should be reasoning

It seems that there is not any figure caption for Figure L141 - L158.

**Details Of Ethics Concerns:**

Medical reasoning data

---

### Official Review · Reviewer_oKTs · 2025-10-26

**Soundness:** 3
**Presentation:** 2
**Contribution:** 3
**Rating:** 6
**Confidence:** 2

**Summary:**

The paper tackles the limited reasoning capabilities of medical visual question answering (VQA) models by combining rationale generation via COMCTS with a self-improvement pipeline.

It first builds a rationale-augmented dataset using strong external VLMs (Gemma-3-27B, Qwen2-VL-7B) and an LLM judge (DeepSeek-R1), then bootstraps a smaller model by learning from its own generated rationales.

A key novelty is filtering self-generated samples using K-medoids clustering over reasoning trajectories, with DTW distances computed from image and sentence embeddings to retain high-quality, on-distribution chains of thought.

Experiments across VQA-RAD, SLAKE, PathVQA, PMC-VQA, and OmniMed-VQA show consistent average gains over baselines, with the CA+CL filter outperforming correctness-only filtering in both accuracy and BLEU-1.

**Strengths:**

Originality: Introducing trajectory-level filtering with K-medoids + DTW for rationale selection in medical VQA is novel and complementary to correctness-based filtering. Leveraging COMCTS to seed high-quality rationales in a low-rationale domain is a creative adaptation of recent reasoning-generation methods.

Quality: The pipeline is clearly modular (dataset construction, self-improvement, filtering, SFT), with ablations across Mbase, Mreas, MSI-ca, and MSI-ca+cl demonstrating incremental benefits. Use of both accuracy and BLEU-1 offers a more nuanced view for short-form answers typical in medical VQA.

Clarity and significance: The method and algorithms are described with sufficient detail (feature extraction, DTW definition, clustering procedure), and figures illustrate the workflow. Given the scarcity of rationale-rich medical data, releasing COMCTS-generated rationales and code could materially benefit the community.

**Weaknesses:**

Limited scale and generality: Due to computational constraints, the study uses restricted subsets of datasets and a fixed K=10 with p=0.2 pruning, which may limit external validity and sensitivity analysis. More thorough scaling experiments and hyperparameter sweeps (K, p, embedding backbones) would strengthen claims.

Dependency on external LLMs/VLMs: The approach relies on stronger proprietary/open models (DeepSeek-R1, Gemma-3-27B, Qwen2-VL-7B) for both generation and judging, raising cost and reproducibility concerns. A discussion of budget, latency, and access constraints, plus open-source alternatives or lighter judges, would be helpful.

Evaluation breadth: While average gains are reported, some per-dataset/model combinations regress (e.g., PathVQA, PMC-VQA in accuracy for certain backbones), and clinical robustness is not assessed. More granular analysis of error types, reasoning faithfulness, and safety/clinical relevance would improve the evaluation.

**Questions:**

How sensitive are the results to the choice of sentence/image encoders and to K and pruning fraction p in K-medoids, and could adaptive clustering (e.g., silhouette- or density-based) yield better filtering?

Can you quantify the end-to-end compute and API cost of COMCTS generation, answer verification, and clustering, and provide guidance for practitioners with constrained budgets?

Do DTW-cluster outliers ever correspond to valid but novel reasoning paths, and if so, could you add a controlled experiment that preserves a subset of outliers to test for creativity vs. noise?

---

### Official Review · Reviewer_NuXQ · 2025-10-30

**Soundness:** 2
**Presentation:** 1
**Contribution:** 1
**Rating:** 2
**Confidence:** 3

**Summary:**

This paper addresses the challenge of limited reasoning capabilities in medical Visual Question Answering (VQA) by proposing a self-improvement framework. The core of the method involves two stages: first, generating reasoning trajectories for existing medical VQA datasets using the off-the-shelf COMCTS algorithm; second, using a model trained on this data to generate more reasoning samples on unlabeled data, which are then filtered based on answer correctness and a novel clustering of reasoning trajectories using K-medoids and Dynamic Time Warping (DTW). The filtered data is used to further train the model. Experiments on five medical VQA datasets show that the proposed filtering strategy can lead to performance improvements over baselines in terms of accuracy and BLEU-1 scores.

**Strengths:**

1. Practical Problem Focus: The work tackles a genuine and important problem in medical VQA: the scarcity of high-quality, rationale-augmented datasets. The effort to generate and release such a dataset is a positive contribution to the community.
2. Comprehensive Evaluation: The evaluation is conducted across multiple established medical VQA datasets (VQA-RAD, Slake, etc.), which adds credibility to the claimed improvements. The use of both exact-match accuracy and BLEU-1 provides a more nuanced view of model performance.

**Weaknesses:**

1. Significant Issues with Motivation and Narrative:

The introduction and related work sections fail to build a compelling narrative. They do not sufficiently clarify the specific limitations of existing medical VQA models and datasets to motivate the need for this particular self-improvement approach. The transition from the general problem to the proposed solution feels abrupt.

The motivation for using reasoning trajectory clustering is particularly weak. The paper lacks a clear, intuitive explanation or preliminary analysis showing that "high-quality" reasoning paths naturally form clusters in the trajectory space, and that noisy paths are outliers. This makes the core filtering mechanism seem like an ad-hoc choice rather than a well-motivated design.

2. Limited Methodological Novelty:

The foundational step of generating reasoning data relies entirely on the COMCTS algorithm, which is not an original contribution of this work. While its application to medical VQA is valid, it sets the baseline for the paper's novelty.
The clustering-based filtering, while novel in this specific context, is presented without strong justification. The idea of filtering data based on feature similarity is well-established. The specific use of DTW on reasoning sequences is interesting but feels incremental without a deeper analysis of why this is the right metric for reasoning quality in medical VQA, or a comparison to simpler semantic similarity measures.

3. Incomplete and Potentially Unreliable Experimental Setup:

As correctly pointed out, the experiments are conducted on random subsets of the original datasets due to computational constraints. This raises serious concerns about the statistical significance and generalizability of the results. The performance on such small subsets may not reflect the true performance on the full datasets.

The results are reported from a single run. Given the randomness in subset selection, model initialization, and the stochastic nature of training, the reported improvements could be due to chance. The community standard for claiming a meaningful improvement is to report the mean and standard deviation across multiple random seeds. The absence of this undermines the credibility of the conclusions.

The performance gains, while positive on average, are inconsistent across datasets and models (e.g., performance drops in several cases for Path-VQA and PMC-VQA). This inconsistency is not adequately discussed or analyzed, leaving the reader uncertain about the method's robustness.

**Questions:**

The reasoning trajectory sequence is defined as $S=(r_0,r_1,…,r_m)$, where $r_0$ is solely the image feature. However, the entire reasoning process is driven and constrained by the "question". Could the authors clarify how the textual information of the question is incorporated into the similarity calculation of these trajectories? If the question features are indeed absent, please justify this design choice, as different questions for the same image should lead to distinct reasoning paths. Neglecting the question may fundamentally undermine the DTW-based clustering approach

---

### Official Review · Reviewer_RKb5 · 2025-10-30

**Soundness:** 1
**Presentation:** 1
**Contribution:** 1
**Rating:** 0
**Confidence:** 4

**Summary:**

The paper fails to adhere to the ICLR author guidelines, most notably by not providing references for the cited literature.

The paper lacks sufficient details regarding the experimental procedures conducted on the VQA-RAD, Slake-VQA, Path-VQA, and PMC-VQA datasets.

The reported accuracy is significantly lower than that in other studies; however, the reviewer cannot fairly compare these metrics because insufficient information is provided.

**Strengths:**

Same as summary

**Weaknesses:**

Same as summary

**Questions:**

Same as summary

---

### Meta-Review · Area_Chair_kSJy · 2026-01-07

**Summary:**

The SAC has indicated a desk rejection and pointed out that *there are no references listed after the main text*.

**Reviewer Concerns:**

Desk rejection flagged by SAC

**Reviewer Scores:**

Desk rejection flagged by SAC

---

### Decision · Program_Chairs · 2026-01-26

Reject